# Effects of Nano-Graphene Oxide on the Growth and Reproductive Dynamics of *Spodoptera frugiperda* Based on an Age-Stage, Two-Sex Life Table

**DOI:** 10.3390/insects13100929

**Published:** 2022-10-13

**Authors:** Cao Li, Chaoxing Hu, Junrui Zhi, Wenbo Yue, Hongbo Li

**Affiliations:** 1The Provincial Key Laboratory for Agricultural Pest Management of the Mountainous Region, Institute of Entomology, Guizhou University, Guiyang 550005, China; 2Institute of Plant Protection, Guizhou Academy of Agricultural Sciences, Guiyang 550006, China

**Keywords:** graphene oxide, age-stage two-sex life table, *Spodoptera frugiperda*, growth and reproduction

## Abstract

**Simple Summary:**

Fall armyworm (FAW), *Spodoptera frugiperda,* is an important pest in a variety of different crops. Graphene oxide (GO) is a promising candidate used in a biological context because of its versatility. In agriculture, GO could be potentially used as a pesticide additive to improve the efficacy of insecticides. In this study, the effects of GO on the development and reproduction of FAWs were determined based on life table analysis. The results showed that GO could prolong the duration of the egg stage and instar larval stages, but shorten lifespan of male and female adults, and this effect was enhanced with increasing GO concentrations. GO also reduced the number of eggs laid by female moths. In addition, the expression of genes related to reproduction have also been affected by GO. In conclusion, GO prolonged the developmental period of FAWs, decreased fecundity, and may decline the population size. The study provides a basis for the rational use of GO as a pesticide synergist for FAW control.

**Abstract:**

The development and reproduction of the fall armyworm (FAW), *Spodoptera frugiperda*, which were reared on artificial diets containing nano-graphene oxide (GO), were determined based on age-stage, two-sex life table analysis. The results showed that GO had adverse effects on FAWs. Compared with the control, the duration of the egg stage and first, second, and sixth instar larval stages increased with increasing GO concentrations; however, the lifespan of male and female adults decreased with increasing GO concentrations. Weights of FAW pupae that were supplied with GO-amended diets increased by 0.17–15.20% compared to the control. Intrinsic growth, limited growth, and net reproductive rates of FAWs feeding on GO supplemented diets were significantly lower than the control, while mean generational periods (0.5 mg/g: 38.47; 1 mg/g: 40.38; 2 mg/g: 38.42) were significantly longer than the control. The expression of genes encoding vitellogenin (*Vg*) and vitellogenin receptor (*VgR*) expression was abnormal in female FAW adults feeding on GO-amended diets; the number of eggs laid decreased relative to the control, but *Vg* expression increased. In conclusion, GO prolonged the developmental period of FAWs, decreased fecundity, and led to a decline in the population size. The study provides a basis for the rational use of GO as a pesticide synergist for FAW control.

## 1. Introduction

Nanomaterials (NMs) are small particles with a large surface area and high reactivity that have different degrees of toxicity toward the environment and living organisms [1,2,3]. Nanomaterials have novel properties, exhibit a broad range of applications [4], and are used in diverse fields including agriculture, cosmetics, nanomedicine, and environmental protection [5]. Graphene oxide (GO) is prepared by oxidating graphite with strong acids and has industrial, environmental, and biotechnological applications [6,7]. Although the chemical property of GO is understood, the role of graphene family materials in biological responses is unclear, and their potential toxicity remains controversial. Some studies have reported that NMs have no significant toxicity or negative biological effects in the cell cultures of various plant and animal species [8,9]. Hu et al. demonstrated that GO induced damage to algal cells, thus resulting in oxidative stress and metabolic disorders [10]. Furthermore, GO was shown to inhibit rice development, thus reducing numbers of roots and the weight of roots and shoot [11]. The sharp edges of GO particles may induce cell membrane damage and induce the formation of reactive oxygen species [12,13].

GO has been used as a pesticide additive to improve the efficacy of insecticides. Sharma and co-workers reported that GO binds pesticides, reduces pesticidal drift and controls pests [14]. Tong et al. demonstrated that nanocomposites consisting of GO and the fungicide hymexazol improved the adsorption and utilization rate of hymexazol by plants [15]. In another study, GO functioned synergistically with pyridine, chlorpyrifos, and permethrin and improved the acaricidal effect of the three pesticides [16]. Short-term exposure of the house cricket, *Acheta domesticus*, to GO in food caused an increase in oxidative stress, induced DNA damage, and caused an increase of apoptosis [17], which shows the potential application of GO for pest control. Major breakthroughs have been made in nanotechnology, and NMs are widely used in a variety of fields [18]. Studies on the implementation of NMs for disease and insect pest control, and assessment of NMs risk on non-target organisms, are clearly needed to successfully deploy nanomaterials in agriculture. Insects are an excellent choice for these studies because of their availability, short reproductive cycles, and large population sizes.

*Spodoptera frugiperda* (J.E. Smith) (Lepidoptera: Noctuidae), commonly known as the fall armyworm (FAW), is an important pest in a variety of different crops [19]. It has strong reproductive ability, high population density, and voracious feeding habits that lead to huge economic losses [20]. FAWs originated in the Americas and spread rapidly throughout Africa and Asia [20,21,22]; it is polyphagous and feeds on more than 353 host plants in the Americas [23]. At present, effective control of FAWs is generally achieved with chemical pesticides; however, the long-term use of chemical pesticides on FAWs will seriously shorten their efficacy due to insecticide resistance [24]. Therefore, more efficient ways of delivering pesticides are needed to reduce resistance and prolong the effectiveness of chemicals used to control FAWs. Therefore, it is important to evaluate the toxic effects of GO on FAW reproduction and development.

In this study, FAWs were allowed to feed on different concentrations of GO, and the effects on FAW growth and population dynamics were analyzed using age-stage, two-sex life tables. In addition, expression of the vitellogenin and vitellogenin receptor genes to GO were determined. The responses of eggs, larvae, prepupae, pupae, male, female adults, and reproduction related genes to GO were evaluated, and the ecotoxicological effects of GO were used to provide a theoretical basis for safe, rational utilization of GO in FAW control.

## 2. Materials and Methods

### 2.1. Insects

The original FAW colony was collected in 2019 from infested maize in Yuxi, Yunnan, China (24°19′ N, 101°16′ E). Larvae were reared on artificial diets at 25 ± 1 °C under a 16:8 h light: dark photoperiod and 70 ± 5% relative humidity.

### 2.2. Preparation of GO Diets

Single-layer GO dispersion tablets (50–200 nm diameter, initial concentration 2 mg/mL) were purchased from Nanjing Xianfeng Nano Material Technology Co., Ltd. (Nanjing, China).

Artificial diets were prepared as described for feed formula II [25] with minor modifications (Table 1).

### 2.3. Characterization and Stability of GO

The single-layer GO dispersion tablets used in this study were sent to Shanghai Huiming Testing Equipment LLC (Shanghai, China) for characterization and stability measurements. Transmission electron microscopy (TEM) was used to observe the ultrastructure of GO samples and the zeta potential was determined to evaluate the stability of GO dispersion.

### 2.4. Developmental Times

GO was added to artificial diets at 0, 0.5, 1, and 2 mg/g and an experimental population life table of FAWs was established to evaluate the effects of GO on the growth, development, and reproduction of FAWs. Newly hatched first instar larvae were transferred to 24-well plates with a soft brush and allowed to feed on diets containing 0 (control), 0.5, 1, and 2 mg/g GO to adulthood. After the females laid eggs, the eggs were dispensed to 24-well plates (2.0 × 2.5 cm wells), and each treatment contained 50 eggs and 3 replicates. The number of hatched larvae was recorded daily, and diets and plates were changed daily. The developmental period (time from egg to pupation), survival rate, time needed for transition from prepupae to pupae, and the weights of three-day-old pupae were recorded. After adult emergence, male and female sexes were distinguished by color and the presence or absence of markings on the forewing. Male and female FAWs were transferred in pairs to 1750 mL plastic boxes containing cotton balls soaked in 10% honey water. Gauze was placed inside the boxes to provide a platform for oviposition. The rearing box was placed in a growth chamber, and cotton balls were replaced every two days. After females laid eggs, the number of eggs and hatchability of each FAW pair was observed and recorded.

### 2.5. Age-Stage, Two-Sex Life Table

The duration of larval development, adult life span, and the fecundity of individual females were determined as described in Section 2.4; these results were used to construct age-stage, two-sex life tables [26,27,28] at the three different concentrations of GO as described previously [29,30]. Age-stage-specific survival rates (*S_xj_*) represent the probability of individual FAWs surviving from newly hatched eggs to age *x* stage *j*, and a survival rate of 0 indicates that the FAWs failed to develop to the next age. Overlaps indicated the presence of generational overlaps in FAW populations. Survival rates were based on the daily survival of FAW populations, and age-specific survival rates (*l_x_*) represent the survival of FAWs from newly hatched eggs to age *x*, regardless of stage differentiation, where *j* represents the developmental stage and
$$S_{xj}=\frac{n_{xj}}{n_{0}}$$
$$l_{x}=\sum_{j=1}^{k}S_{xj}$$*n*_0_ is the number of eggs (newborns) at the beginning of the life table study.

Age-stage-specific fertility (*f_xi_*) is the number of hatched eggs produced by female adults at age *x* and age-specific fecundity (*m_x_*) is the number of eggs per individual at age *x*.
$$m_{x}=\sum_{j=1}^{k}S_{xj}f_{xj}/\sum_{j=1}^{k}S_{xj}$$

Age-specific maternity (*l_x_*×*_x_*) is the product of *l_x_* and *m_x_*, and age-stage-specific life expectancy (*e_xj_*) is the time that an individual of age *x* and stage *y* is expected to live.
$$e_{xj}=\sum_{i=x}^{a}\sum_{y=j}^{k}{S^{\prime }}_{iy}$$

*S*′*_iy_* is the probability that an individual of age *x* stage *y* survives to age *i* stage *j*, and the age-stage-specific reproductive value (*V_xj_*) is the contribution of individuals of age *x* and stage *y* to future populations:$$v_{xj}=\frac{e^{r\left(x+1\right)}}{S_{xj}}\sum_{i=x}^{a}e^{-r\left(i+1\right)}\sum_{j=1}^{k}{S^{\prime }}_{iy}{f^{\prime }}_{iy}$$

The formula for population parameters is as follows:

*r*, intrinsic rate of increase:$$\sum_{x=0}^{\infty }\;e^{-r\left(x+1\right)}l_{x}m_{x}=1$$

*λ*, finite rate of increase:λ = e*^r^*

*R*_0_, net reproductive rate:$$R_{0}=\sum_{x=0}^{\infty }l_{x}m_{x}$$

*T*, mean generation time:*T* = (ln *R*_0_)/*r*

### 2.6. Gene Expression Studies

Based on ovarian development and prior grading of FAWs [31], female adults that were reared on different concentrations of GO (control, 0.5, 1, 2 mg/g) were sampled at 1, 3, 6, and 11 d after eclosion. Three biological replicates were set for each sampling, and each replicate includes five female adults. After sampling, female adults were placed on ice for 3–5 min, wiped with 70% alcohol for 30 s, soaked in 0.25% in sodium hypochlorite for 1 min, and then rinsed in sterile water three times to remove external contaminants. Fine-tipped, sterile forceps were used excise ovaries, which were immediately rinsed in 0.9% sterile NaCl, washed twice and placed in individual 1.5 mL microcentrifuge tubes. The samples were flash-frozen in liquid nitrogen and placed in a −80 °C freezer until needed.

The total RNA of the sample was extracted with the Eastep SuperTotal RNA Extraction Kit (Shanghai Promega Biological Co., Ltd. (Shanghai, China)). RNA quality was evaluated by 1% agarose gel electrophoresis, and the concentration and purity of RNA were detected by spectrophotometry (Nano Photometer P-Class). RNA templates (2 μL) were removed, and cDNA was synthesized using the HiFiScript cDNA Synthesis Kit (Beijing Kangwei Century Biotechnology Co., Ltd. (Beijing, China)). Total RNA and cDNA templates were stored at −80 and −20 °C, respectively.

RT-qPCR primers were designed by Primer Premier v. 6.0 using a conserved sequence of the *Vg* gene released by NCBI (Table 2). *RPL27* encoding ribosomal protein L27 (GenBank accession no. AF400191. 1) was used as an internal reference gene, determining the vitellogenin gene (*SfVg*) and vitellogenin receptor gene (*SfVgR*) in FAWs. FastStart Essential DNA Green Master Mix (Roche Diagnostics GmbH, Penzberg, Germany) was used for quantification of fluorescence. The total reaction volume of 20 μL contained the following reagents: cDNA template, 2 μL; ddH_2_O, 6 μL; and 1 μL each of the upstream and downstream primers (10 μmol/L). The RT-qPCR procedure consisted of the following steps: pre-denaturation at 95 °C for 10 min; denaturation at 95 °C for 30 s, annealing at 56 °C for 30 s, 40 cycles; 65 °C for 5 s, and then 95 °C for 5 s.

### 2.7. Data Analysis

The life parameters of the FAWs were calculated using the TWOSEX-MS Chart 2020 program [32]. The mean and standard error of each parameter were evaluated by bootstrapping with 100,000 replicates [33], and significant differences were identified using the paired bootstrap test [34]. Relative gene expression was calculated using the 2^−ΔΔCt^ quantitative method. SPSS v. 24.0 (SPSS Inc., Chicago, IL, USA) was used to conduct one-way ANOVA for *SfVg* and *SfVgR* data, and Tukey’s HSD method was to determine significant differences. Relationships between stage-specific oviposition quantities, stage-specific hatching rates, and gene expression were determined by Pearson correlation analysis.

## 3. Results

### 3.1. Characteristics of GO

The sheet diameters of GO samples were between 50 and 200 nm (Figure 1), and the potential of the dispersion was ±50 (Figure 2), which indicated that the GO material was a stable nanoscale GO dispersion.

### 3.2. Developmental Duration of FAW Life Stages

The effects of different GO concentrations on the developmental stages of FAWs are shown (Table 3). Compared with the control, the developmental duration of the egg stage, first, second, and sixth instars, and the prepupal stage of FAWs fed on the GO diet was significantly prolonged at increasing GO concentrations. Conversely, the developmental duration of the fourth instar larvae decreased with increasing concentrations. When the GO concentration was 0.5 mg/g, the larval developmental period was 20.20 d compared to the control at 19.67 d. At the highest dose of 2 mg/g, the larval stage was 20.36 d, which was 0.69 d longer than the normal control diet. The immature period gradually increased with increasing GO concentrations and was longest (32.47 d) when the FAWs were fed 1 mg/g GO; this duration was significantly longer than the control.

### 3.3. FAW Pre-Oviposition, Egg Production, Pupal Weight, Female-to-Male Ratio and Lifespan

There were also differences in GO effects on the lifespan of FAW adults (Table 4). When the concentration of GO was 2 mg/g, the female adult lifespan was 12.48 d, which was significantly decreased relative to the control at 16.04 d. Furthermore, the female adult lifespan at 2 mg/g GO was significantly shorter than the lifespan at 0.5 mg/g (14.62 d) and 1 mg/g (15.64 d) (*p* < 0.05,). After feeding different GO concentrations, the lifespan of male adults was also reduced and was the shortest at 1 mg/g (11.65 d). However, there was no significant difference in the lifespan of FAW male adults at different concentrations (*p* > 0.05). The total lifespan of FAWs was shortened by 2.12 d compared with the normal diet, and the average number of eggs laid was reduced by 1.81 times compared with the control (normal diet). Similarly, FAW pupal weights decreased in response to GO treatments, and this was concentration-dependent; for example, pupal weight at 2 mg/g was 193.6 mg, which was significantly lower than the control at 228.29 mg (Table 4). There was also a significant increase of the female/male ratio after feeding on GO at 2 mg/g (*p* < 0.05). Significant differences were also observed among treatments in the adult pre-oviposition period; for example, the pre-oviposition stage was longest (6.89 d) when the GO concentration was 1 mg/g (Table 4).

### 3.4. Survival Rates of FAWs

Differences were observed in the age-stage survival rates of FAWs after feeding on three concentrations of GO (Figure 3). The survival curves for different FAW life stages overlapped after GO feeding, indicating that different insect stages exist at the same time due to variations in development, thus resulting in overlapping generations. The *S_xj_* curves showed that the survival rates of female adults fed at 0.5 mg/g and 2 mg/g GO were higher than male adults. Differences also existed in the survival rate of FAW larvae fed with three GO concentrations (Table 5). After feeding on a GO-amended diet, the survival rate of larvae was significantly different at the egg, fourth and sixth instar, and pupal stages. The survival rate of the egg stage was significantly lower than the control, and the survival rate of the pupal stage was lowest after feeding on 2 mg/g GO; the latter results indicated that high concentrations of GO reduced FAW survival at the pupal stage.

### 3.5. Fecundity of FAWs

The results for the population-specific age-specific survival parameter *l_x_* is shown in Figure 4. The *l_x_* statistic at three GO concentrations showed a steep decline in early stages and a steady decline in later stages, and the change trend of the *l_x_* curve was similar in the early stages. When developmental duration parameters and the *S_xj_* curve were considered together, the data indicated that FAWs had a higher mortality risk in the larval and pupal stages after feeding on the GO diet. At different GO concentrations, the age-stage fecundity parameter *m_x_* increased and then decreased, and fecundity was the lowest at 2 mg/g. GO concentrations of 1 and 2 mg/g resulted in lower survival than GO at 0.5 mg/g and the control for *m_x_*, *l_x_ m_x_* and *f_x_*. The results indicated that the higher GO concentrations in the diet were not conducive to the growth, development, and reproduction of FAWs.

### 3.6. Life Expectancy of FAWs

Age-specific life expectancy (*e_xj_*) represents the amount of time an individual at age *x* age *j* was expected to live. The life expectancy curves of FAWs fed on the three concentrations of GO showed a slow decline (Figure 5). Among these, the highest life expectancy was at *e*_01_, and the maximum life expectancy of FAWs fed on GO diets was lower than the control (35.71 d).

### 3.7. Reproductive Value of FAWs

At GO concentrations of 0.5, 1, and 2 mg/g, the reproductive values of FAWs at the beginning of oviposition were 1.15, 1.14, and 1.14 d, respectively; these values were significantly lower than the control (1.16 d) and similar to the cycle-limited growth rate of the population life table parameter. The reproductive value of FAWs at different concentrations and stages increased with the maturation of instars and was highest in the adult stage, which indicated that the adult stage was the most important predictor of future populations. When GO concentrations were 0.5, 1, and 2 mg/g, the maximum age-stage reproductive values of female adults were 516.74, 506.03, and 372.36 offspring/adult, respectively; these values were lower than the control (598.42 offspring/adult) and peaked at 35 d, 36 d, and 35 d, respectively (Figure 6).

### 3.8. Life Table Parameters

Intrinsic growth, cycle growth, and net reproductive rates of FAWs decreased as GO concentrations increased (Table 6). The intrinsic and net reproductive rates were lowest at 2 mg/g GO and were 1.14 d^−1^ and 170.36 eggs, respectively. Intrinsic growth, weekly growth, and the net reproductive rates were highest in the control group and were 0.154, 1.16 d^−1^, and 329.37 eggs, respectively. In contrast, the average generational period was highest at 1 mg/g GO at 40.38 d, and this period was significantly higher than the other two GO concentrations and the control (Table 6).

### 3.9. Expression of SfVg and SfVgR

After feeding on GO-supplemented diets, *SfVg* expression varied in female FAW adults at different time periods (Figure 7). The relative expression of *SfVg* increased after adult emergence, peaked on day three and then declined significantly. On day one, *SfVg* expression was highest when diets were supplemented with 0.5 mg/g GO and lowest at the 2 mg/g GO concentration. The highest *SfVg* expression levels in female adults were observed with 2 mg/g GO at day three; this level was significantly higher than the 0.5 and 1 mg/g concentrations and was higher than the control. On day six, *SfVg* expression levels were significantly higher in female adults feeding on 2 mg/g GO and were 3.92-, 2.97-, and 3.74-fold higher than expression levels at 0, 0.5, and 1 mg/g GO, respectively. The overall expression level decreased after 11 d, and there was no significant difference between GO concentrations at this time point.

The relative expression of *SfVgR* in female FAW adults varied at different times (Figure 8). On day one, *SfVgR* expression was significantly higher at 2 mg/g and 0.5 mg/g GO, and expression at the other concentrations was not different from the control. On day three, *SfVgR* expression in the 0.5 and 2 mg/g GO diets decreased relative to the control, and there was significant difference between the decreased levels. *SfVgR* expression at 1 mg/g GO showed an increase relative to the control but was significantly different from the control. *SfVgR* expression in adults fed on 1 mg/g GO was 2.69- and 2.52-fold higher than expression levels at 0.5 and 2 mg/g. On day six, *SfVgR* expression in adults fed on GO was significantly lower than the control, and relative expression levels at 0.5, 1, and 2 mg/g GO were 2.51-, 3.86-, and 1.72-fold lower than the control, respectively. On day eleven, *SfVgR* expression in FAWs feeding on 0.5 mg/g and 1 mg/g GO was 1.89- and 3.94-fold higher than the control.

### 3.10. Correlation Analysis of Gene Expression Levels and Fecundity

The total oviposition period for FAWs after feeding on GO-amended diets varied with the GO concentration; when GO was added to diets at 0.5, 1, and 2 mg/g, the oviposition period decreased by 2, 3, and 6 d, respectively (Table 7). In the third stage, there was a significant decrease of the oviposition quantity in FAWs feeding on 2 mg/g GO, and the quantity on the 2 mg/g GO diet was reduced 2.8-fold compared to the control. In contrast, the hatching rate of different stages decreased by 0.04-fold after the FAWs were supplied with 2 mg/g concentrations of GO in the fourth stage.

Pearson-derived correlation coefficients derived for *SfVg* expression and FAW egg production at different concentrations of GO were between 0 and 0.01, indicating that *SfVg* expression and egg production were not correlated (*p* > 0.05).

## 4. Discussion

In this study, age-stage, two-sex life tables for FAWs at different GO concentrations were constructed based on laboratory experiments and the toxicological effects of GO on the growth, development, and reproduction of FAWs were determined. The results showed that GO inhibited FAW growth and development, decreased pupal weights, and reduced egg production. We also observed deviations in *SfVg* and *SfVgR* expression in FAW adult females consuming GO diets. Our findings further illustrate the toxic effects of GO on the reproduction and development of lepidopteran insects, which is consistent with a prior study that focused on the deleterious effects of GO on FAW larvae [35]. However, some variations exist between our results and those reported by Martins et al. [35]; this could be due to the different dimensions of the GO platform and/or the different formulations of GO (solid vs. dispersive liquid in the present study). The toxicity of different nanomaterials to insects depends on the type and concentration of nanomaterials, insect species, handling procedures for NMs, and absorption by insects [36]. For example, silver nanoparticles stunted the growth of *Heliothis virescens* and *Trichoplusia ni*, reduced pupal and adult weights, and decreased fecundity [37]. In another study, 20 mg L^−1^ nanosilver significantly increased the body weight of silkworms and had no negative effects on survival but reduced cocooning rates in a dose-dependent manner [38].

The growth rate of lepidopteran larvae depends on the efficient acquisition and utilization of essential nutrients in the diet. Food quality plays a central role in insect life history, physiology, and biochemistry and strongly influences trophic interactions [39]. Our results showed that GO treatment resulted in reduced pupal weights, possibly because GO interfered with nutrient uptake and resulted in disordered, abnormal development. Panacek et al. reported that silver NPs were toxic to *Drosophila melanogaster* and caused a significant reduction in lifespan and fertility [40]. Furthermore, Ibrahim et al. reported that silver and zinc oxide nanoparticles inhibited the weight of *Spodoptera littoralis* larvae and pupae [41], possibly because nanomaterials affect nutrient acquisition and utilization in insects.

Nanomaterials vary considerably with respect to their effects on insects. For example, fullerene Cbo, carbon black, and single- or multi-wall nanotubes were added to *D. melanogaster* diets and had no effect on growth [42]. Similarly, carbon nanotubes had no significant effect on the fecundity or fertility of FAWs [35]. Furthermore, GO had no significant effect on the mortality of Asian corn borer or spider mites during a 24 h monitoring period [16]. Different concentrations of nano-CeO_2_ had no significant effects on the fecundity or mortality of *Myzus persicae* [43]. However, GO promoted the growth and development of Asian corn borer and enhanced pupal weights [44]. In contrast, our study shows that different GO concentrations had negative effects on the growth, development, and reproduction of FAW, and GO at 2 mg/g significantly reduced the survival rate of FAWs at the pupal stage. When adult zebrafish were treated with nano-titanium oxide (nano-TiO_2_) at 100 μg/L, the permeability of the intestinal epithelial barrier was perturbed, thus causing inflammation and oxidative stress [45]. It is also possible that because of the enrichment effect of the existence of NMs, the enrichment amount reaches the maximum at the pupal stage, which eventually leads to a decrease of the pupal weight and an increase in the mortality of the pupal stage. For example, gold nanomaterials were shown to accumulate in both tobacco leaves and *Manduca sexta* larvae after feeding [46]. After ingesting nanomaterials, organisms undergo physiological and biochemical reactions, and the response time, mode, and degree vary according to the biological species.

In this study, age-stage, two-sex life table analysis was used to evaluate the effect of GO on FAW populations. GO had significant effects on larval developmental time and FAW fecundity. High concentrations of GO significantly reduced the fecundity and reproductive parameters of FAWs. Other studies have also shown that diets containing high concentrations of carbon nanomaterials significantly reduce FAW fecundity and fertility [35]. GO was also shown to reduce the reproductive capacity of *Acheta domesticus* in a concentration-dependent manner [47]. Similarly, we show that the FAW population gradually decreased after GO treatment compared to the control. As the GO concentration increased, FAW fertility decreased; thus, GO has toxic effects on the growth, development, and reproduction of FAWs.

Vitellogenin and the vitellogenin receptor play important roles in the maturation of insect ovaries [48] and are critical to insect fertility. After FAWs were allowed to feed on GO, *SfVg* and *SfVgR* showed abnormal expression levels compared to the untreated control, and there were time and concentration effects. On day three, the 2 mg/g GO treatment significantly increased *SfVg* expression, whereas *SfVgR* showed a significant decrease. Previous studies have shown that nanomaterials accumulate in living organisms, especially in reproductive systems where NMs may affect reproduction and development. When nano-TiO_2_ at 100 μg/L was applied to adult zebrafish, the NM particles were transferred to the gonads, resulting in abnormal levels of reproductive hormones, vitellogenin and other indicators [49]. In the present study, changes in GO concentrations impacted *SfVg* and *SfVgR* expression, which ultimately led to reproductive disorders and reduced fecundity. Panacek’s study showed that silver NPs were toxic to *D. melanogaster*, which resulted in the overexpression of stress proteins in vivo [40]. Furthermore, when nano-TiO_2_ was ingested by zebrafish, there was a decrease of the concentration of luteinizing hormones and a significant increase of the concentration of vitellogenin, which resulted in reproductive disorders [50]. GO nanoparticles can accumulate in target organs, leading to the disruption of gut and gonad tissues, which can negatively impact reproductive performance [17]. Kotil et al. reported that increased concentrations of nano-TiO_2_ resulted in autophagy and necrosis of interstitial cells, spermatogenic cells, and spermatogonia, which led to reproductive toxicity [50].

In recent years, numerous studies have been conducted to find out the potential toxicity of graphene family materials (GFMs) on different organisms when assessing the application prospects of GFMs. As stated by Jastrzebska and Olszyna, considering the increase in the use and production of GFMs and the consequent environmental emissions, their toxic effects are becoming an urgent issue [18]. Some studies have shown that GFMs have certain effects on aquatic environments and soil organisms. Xie et al. studied the toxicity of GO on white rot fungus and found that the growth of white rot fungi could be stimulated at a low concentration of GO and be inactivated at a high concentration [51]. Li et al. studied the potential adverse effects of graphite, and graphene quantum dots (GQDs) on the motor nervous system using nematode Caenorhabditis elegans and found that graphene-based nanomaterials could cause damage to the dopaminergic and glutamatergic neurons [52]. Chen et al. showed that GO accumulations occurred in the liver and intestine and caused obvious chronic toxicity to these organs after 14 days exposure [53]. Although the oxicological effects of GFMs to some organisms were determined to a certain extent, much more attention should be paid to a variety of other organisms.

This study analyzed the effects of GO on FAW growth, development, and reproduction in the context of the age-stage life table of both sexes. This study provides a basis for further exploration of the ecotoxicological effects of GO on lepidopteran species and provides a theoretical basis for the rational utilization and deployment of GO for pest control. However, the underlying mechanism of GO toxicity for FAWs remains unknown and further research is warranted. Furthermore, this study focused solely on *SfVg* and *SfVgR* expression, and other genes related to reproduction warrant further study to better understand GO toxicity in FAWs. In addition, whether GO has potential risk to the environment remains to be further studied.

## 5. Conclusions

The development and reproduction of the FAWs that were reared on artificial diets containing GO were determined based on age-stage, two-sex life table analysis. The results showed that GO had adverse effects on FAWs. Compared with the control, the duration of the egg stage and instar larval stages increased with increasing GO concentrations; however, the lifespan of male and female adults decreased with increasing GO concentrations. Weights of FAW pupae that were supplied with GO-amended diets increased compared to the control. Intrinsic growth, limited growth, and net reproductive rates of FAWs feeding on GO supplemented diets were significantly lower than the control, while mean generational periods were significantly longer than the control. Expression of genes encoding vitellogenin (*Vg*) and vitellogenin receptor (*VgR*) expression was abnormal in female FAW adults feeding on GO-amended diets; the number of eggs laid decreased relative to the control, but *Vg* expression increased. In conclusion, GO prolonged the developmental period of FAWs, decreased fecundity, and led to a decline in the population size. The study provides a basis for the rational use of GO as a pesticide synergist for FAW control.

## Figures and Tables

**Figure 1 insects-13-00929-f001:**
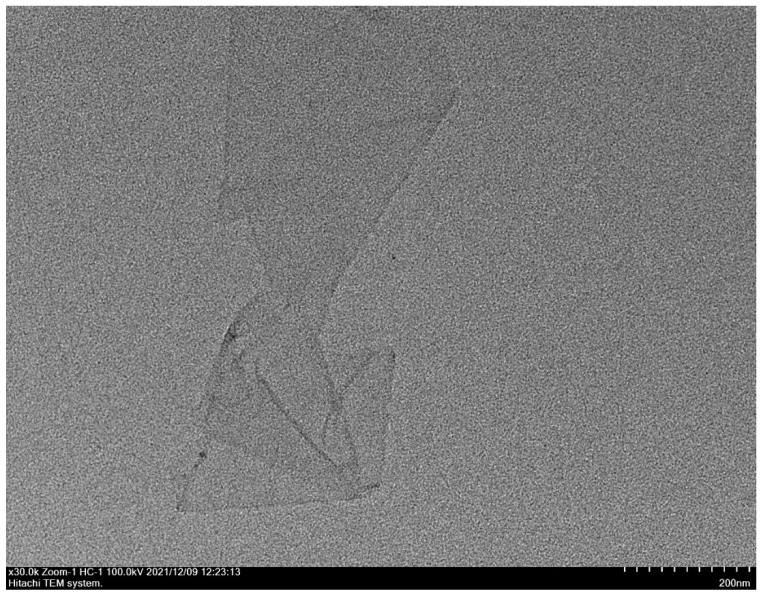
Transmission electron micrograph showing GO dispersion.

**Figure 2 insects-13-00929-f002:**
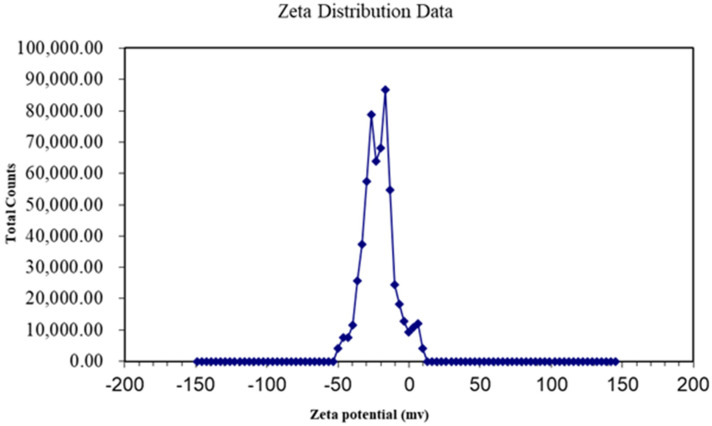
Zeta potential of GO dispersion.

**Figure 3 insects-13-00929-f003:**
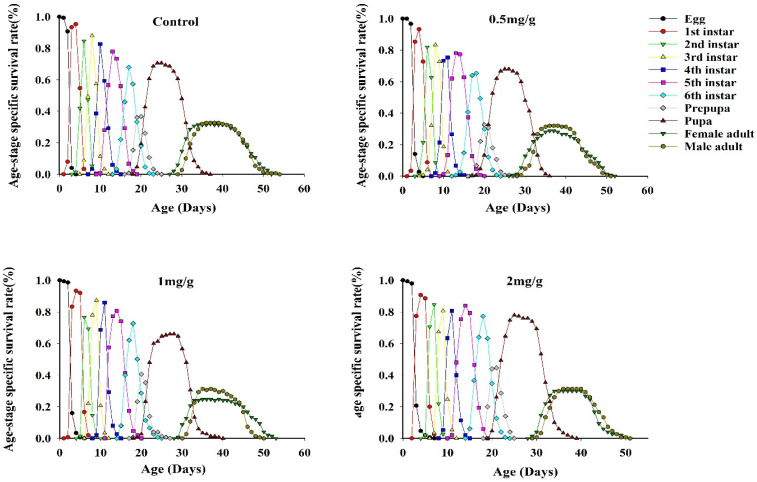
Age-stage-specific survival rates of *Spodoptera frugiperda* at different GO concentrations.

**Figure 4 insects-13-00929-f004:**
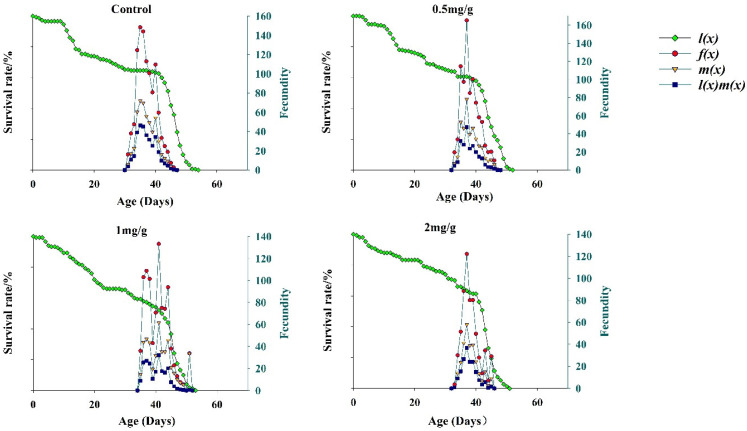
Population age-specific survival and fecundity of *Spodoptera frugiperda* at different GO concentrations.

**Figure 5 insects-13-00929-f005:**
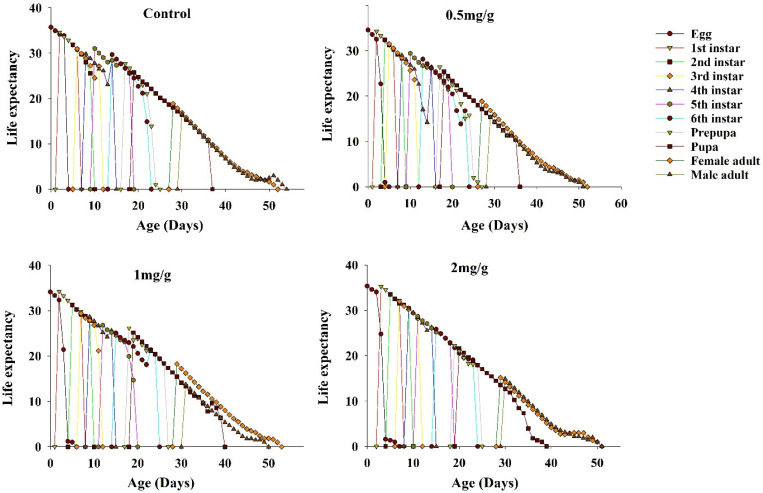
Age-stage specific life expectancy (*e_xj_*) of *Spodoptera frugiperda* at different GO concentrations.

**Figure 6 insects-13-00929-f006:**
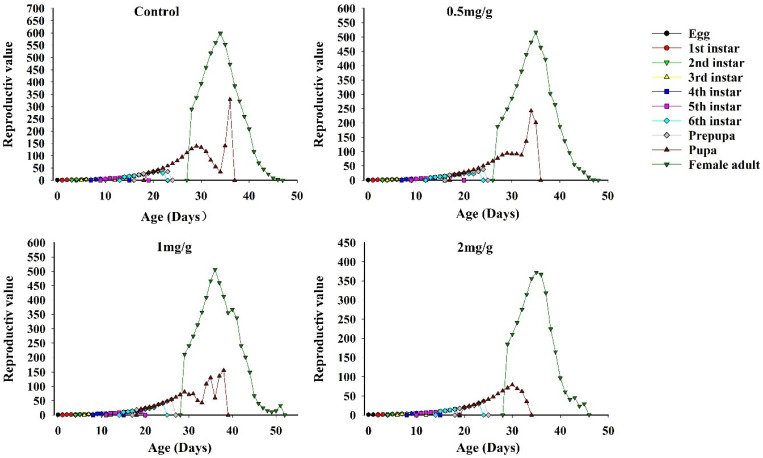
Age-stage characteristic reproductive values *v_xj_* of *Spodoptera frugiperda* at different GO concentrations.

**Figure 7 insects-13-00929-f007:**
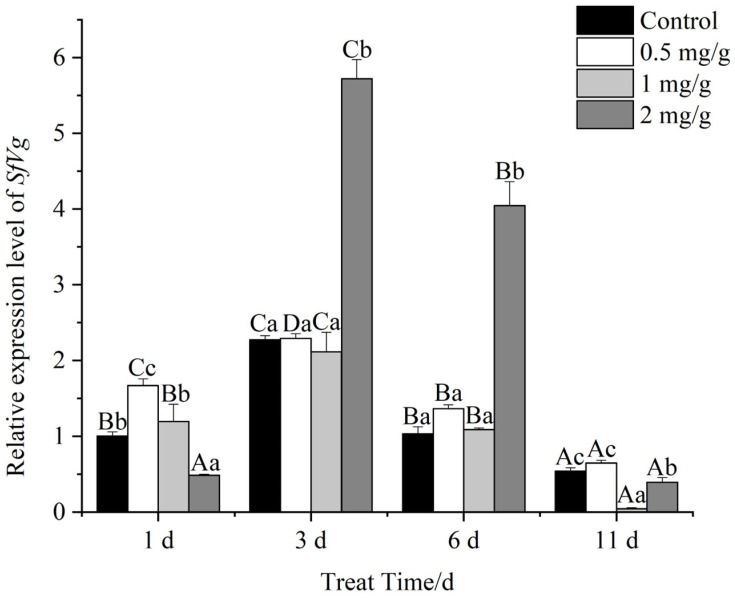
Relative *SfVg* expression levels in female adults of *Spodoptera frugiperda* at different developmental stages. Data are mean ± standard error. Different lowercase letters indicate that the expression levels of genes treated with different concentrations of GO at the same time are significantly different at *p* < 0.05 level as tested by Tukey’s HSD method. Different capital letters indicate highly significant differences in gene expression at different times when treated with the same concentration of GO at *p* < 0.01 level as tested by Tukey’s HSD method.

**Figure 8 insects-13-00929-f008:**
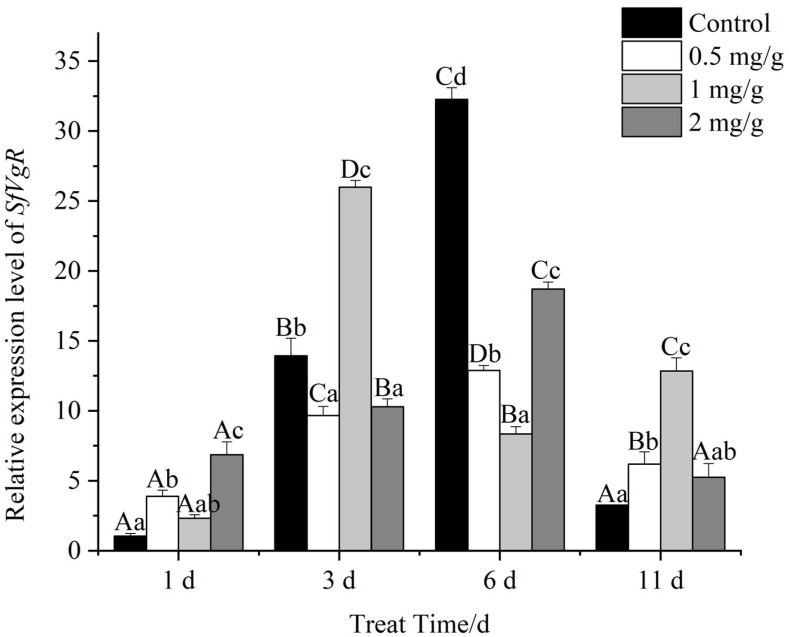
Relative expression levels of *SfVgR* in female adults of *Spodoptera frugiperda* at different developmental stages. Data are mean ± standard error. Different lowercase letters indicate that the expression levels of genes treated with different concentrations of GO at the same time are significantly different at *p* < 0.05 level as tested by Tukey’s HSD method. Different capital letters indicate highly significant differences in gene expression at different times when treated with the same concentration of GO at *p* < 0.01 level as tested by Tukey’s HSD method.

**Table 1 insects-13-00929-t001:** Artificial diet for *Spodoptera frugiperda*.

Ingredients	Ingredient Amount
Distilled water	1200 mL
Agar	24 g
Bean powder	120 g
Wheat bran	120 g
Yeast powder	48 g
Casein	24 g
Sorbic acid	2.4 g
Cholesterol	0.24 g
Inositol	0.24 g
Ascorbic acid	9.6 g
Choline chloride	1.2 g

**Table 2 insects-13-00929-t002:** Primers for RT-qPCR.

Gene	Primer (5′-3′)
*RPL27-F*	GAAGCCAGGTAAAGTGGTGCT
*RPL27-R*	GTGTCCGTAGGGCTTGTCTG
*SfVg-F*	TCCTCAGTGTTAACGTGCCC
*SfVg-R*	ACAGTCCCTGTTCACGTTCC
*SfVgR-F*	GTCGTGTGGTGGAAGCTGTA
*SfVgR-R*	GCCACCAACATTCTCCCGTA

**Table 3 insects-13-00929-t003:** Developmental stages of *Spodoptera frugiperda* fed with different GO concentrations.

Treatment	Larval Stadium/d									Pre-Adult Time/d
	Egg	1st Instar	2nd Instar	3rd Instar	4th Instar	5th Instar	6th Instar	Prepupal Period	Pupae	
Control	2.95 ± 0.02 c	2.62 ± 0.04 c	1.87 ± 0.03 c	2.24 ± 0.03 a	2.44 ± 0.05 a	4.16 ± 0.06 a	3.39 ± 0.06 b	1.74 ± 0.04 ab	10.39 ± 0.10 a	31.59 ± 0.17 c
0.5 mg/g	3.06 ± 0.03 b	2.78 ± 0.02 b	1.85 ± 0.03 bc	2.27 ± 0.04 a	2.21 ± 0.04 b	4.35 ± 0.07 a	3.68 ± 0.06 a	1.59 ± 0.05 b	10.44 ± 0.10 a	31.91 ± 0.19 bc
1 mg/g	3.10 ± 0.02 b	3.08 ± 0.02 a	1.76 ± 0.04 b	2.31 ± 0.03 a	2.23 ± 0.04 b	4.28 ± 0.07 a	3.66 ± 0.07 a	1.79 ± 0.04 a	10.51 ± 0.10 a	32.47 ± 0.19 a
2 mg/g	3.15 ± 0.03 a	3.07 ± 0.02 a	2.01 ± 0.01 a	2.03 ± 0.03 a	2.29 ± 0.04 b	4.22 ± 0.06 a	3.59 ± 0.06 a	1.84 ± 0.04 a	10.36 ± 0.09 a	32.38 ± 0.15 ab

Data are the mean ± standard error. Means in a column followed by different letters were significantly different (Tukey HSD, *p* < 0.05).

**Table 4 insects-13-00929-t004:** Differences in the pre-oviposition period, female egg production, pupal weights, female-to-male ratios, and lifespan of *Spodoptera frugiperda*.

Treatment	Male Adult Longevity (d)	Female Adult Longevity (d)	Adult Pre-Oviposition Period (d)	Fecundity (Eggs /Female)	Total Life (d)	Pupal Weight (mg)	Sex Ratio (Female/Male)
Control	14.12 ± 0.38 a	16.04 ± 0.48 a	4.05 ± 0.15 c	1029.29 ± 101.81 a	46.67 ± 0.28 a	228.29 ± 2.27 a	0.95:1 b
0.5 mg/g	12.03 ± 0.49 b	14.62 ± 0.64 a	5.41 ± 0.38 b	810.60 ± 107.08 ab	45.14 ± 0.38 bc	227.91 ± 3.31 a	1.03:1 ab
1 mg/g	11.65 ± 0.42 bc	15.64 ± 0.67 a	6.89 ± 0.60 a	823.28 ± 102.97 a	45.89 ± 0.37 ab	220.45 ± 3.56 a	0.94:1 b
2 mg/g	11.87 ± 0.35 bc	12.48 ± 0.34 b	5.26 ± 0.30 b	567.88 ± 73.14 b	44.55 ± 0.24 c	193.62 ± 2.97 b	1.23:1 a

Data are the mean ± standard error. Means in a column followed by different letters were significantly different (Tukey HSD, *p* < 0.05).

**Table 5 insects-13-00929-t005:** Survival rate at each life stage of *Spodoptera frugiperda* fed on GO.

Survival Rate %	Life Stage								
Treatment	Egg	1st Instar	2nd Instar	3rd Instar	4th Instar	5th Instar	6th Instar	Prepupae	Pupal
Control	97.33 ± 0.01 c	99.32 ± 0.01 a	100.00 ± 0.00 a	96.56 ± 0.02 a	88.72 ± 0.04 a	91.04 ± 0.02 a	97.46 ± 0.01 b	97.35 ± 0.01 a	90.46 ± 0.02 b
0.5 mg/g	94.67 ± 0.01 b	100.00 ± 0.00 a	99.29 ± 0.01 a	98.61 ± 0.01 a	89.93 ± 0.01 a	93.61 ± 0.02 a	95.70 ± 0.02 b	93.9 ± 0.03 a	90.47 ± 0.01 b
1 mg/g	93.33 ± 0.01 ab	100.00 ± 0.00 a	99.29 ± 0.01 a	95.68 ± 0.01 a	95.48 ± 0.01 ab	93.70 ± 0.02 a	88.21 ± 0.01 a	96.12 ± 0.02 a	87.92 ± 0.02 b
2 mg/g	91.33 ± 0.01 a	99.26 ± 0.01 a	98.51 ± 0.01 a	98.48 ± 0.01 a	98.48 ± 0.01 b	96.19 ± 0.01 a	100.00 ± 0.00 b	95.97 ± 0.01 a	78.29 ± 0.01 a

Data are the mean ± standard error. Means in a column followed by different letters are significantly different (Tukey HSD, *p* < 0.05).

**Table 6 insects-13-00929-t006:** Population parameters of *Spodoptera frugiperda* fed with different GO concentrations.

Treatment	*λ* Intrinsic Rate/(d^−1^)	*r* Finite Rate/(d^−1^)	*R*_0_ Net Reproduction Rate/Offspring	T(d) Mean Generation Time/d
Control	1.16 ± 0.01 a	0.15 ± 0.01 a	329.37 ± 50.67 a	37.41 ± 0.30 c
0.5 mg/g	1.15 ± 0.01 ab	0.14 ± 0.01 ab	232.37 ± 42.63 ab	38.47 ± 0.35 b
1 mg/g	1.14 ± 0.01 b	0.13 ± 0.01 b	214.05 ± 39.62 b	40.38 ± 0.43 a
2 mg/g	1.14 ± 0.01 b	0.13 ± 0.01 b	170.36 ± 30.39 b	38.42 ± 0.31 b

Data are the mean ± standard error. Means in a column followed by different letters are significantly different (Tukey HSD, *p* < 0.05).

**Table 7 insects-13-00929-t007:** Total oviposition period and fecundity of *Spodoptera frugiperda* fed with different GO concentrations.

Treat	Total Oviposition Period (d)	Stage Oviposition Quantity				Stage Hatchability			
		Stage 1	Stage 2	Stage 3	Stage 4	Stage 1	Stage 2	Stage 3	Stage 4
Control	18	1495.33 ± 937.79 a	7488.33 ± 2319.71 a	4163.33 ± 613.10 b	2076.66 ± 1054.22 a	0.97 ± 0.06 a	0.98 ± 0.01 a	0.98 ± 0.01 a	0.97 ± 0.01 b
0.5 mg/g	16	3003.00 ± 665.98 a	4262.66 ± 2092.05 a	2628.66 ± 566.35 ab	1837.66 ± 438.11 a	0.95 ± 0.04 a	0.97 ± 0.01 a	0.98 ± 0.01 a	0.96 ± 0.01 b
1 mg/g	15	3703.66 ± 1112.34 a	2204.00 ± 805.66 a	3136.66 ± 621.52 ab	1475.33 ± 588.77 a	0.96 ± 0.02 a	0.98 ± 0.01 a	0.97 ± 0.01 a	0.95 ± 0.01 ab
2 mg/g	12	2193.00 ± 1159.23 a	4386.00 ± 826.18 a	1483.00 ± 581.13 a	449.66 ± 135.47 a	0.96 ± 0.05 a	0.97 ± 0.01 a	0.97 ± 0.01 a	0.93 ± 0.01 a

Total oviposition period was averagely divided into four stages. Data are the mean ± standard error. Means in a column followed by different letters are significantly different (Tukey HSD, *p* < 0.05).

## Data Availability

Data are available upon request from the authors.

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
