# Peer review of "Effects of Nano-Graphene Oxide on the Growth and Reproductive Dynamics of Spodoptera frugiperda Based on an Age-Stage, Two-Sex Life Table"

_insects, 2022, doi:10.3390/insects13100929_

Round 1

Reviewer 1 Report

The study on Effects of nano-graphene oxide on the growth and reproductive dynamics of Spodoptera frugiperda based on an age-stage, two- sex life table. The materials and methods, analysis, are explained clearly and at a reasonable level of detail. I have some comments that need to be addressed to help with the presentation and clarity. Therefore, I think that the manuscript is worthy of publication.

1.       I just only see vitellogenin (Vg) and vitellogenin receptor in abstract, but not in 1. Introduction section, and not in 2.6 Gen expression studies section, I have no idea why test it. So, I suggest add “Expression of genes encoding vitellogenin (Vg) and vitellogenin receptor (VgR) expression” in Introduction.

2.       Table 2I think it is better to note the gene full name.

3.       Table 5, I think instar is not appropriate, because it including egg, larvae, and pupal. Life stage will be better in column.

4.       Line 263-264, heads/adult is not right.

5.       Table 6, please modify the “ Intrinsic rate/(d-1) and Finite rate/d-1” in the table.

6.       Line 274. “1.16 d- and”

7.       Line 279 Are there units for expression levels?

8.       Line 231. Change “Treat” into “Treatment”.

9.       Line 487 Spodoptera frugiperda should be italics

Author Response

Dear Insects Editorial Office

Re: [Insects] Manuscript ID: insects-1926240 - Major Revisions by 7 October 2022

Thank for editors and reviewers’ comments concerning our manuscript entitle ‘Effects of nano-graphene oxide on the growth and reproductive dynamics of Spodoptera frugiperda based on an age-stage, two-sex life table’.

Thank for you and the reviewers’ valuable comments on the improvement of our manuscript. We are very grateful for your detailed instructions on the revision and submission process of our manuscript.

We have carefully considered each comment from the reviewers. We have done the corresponding changes in the 'Revision' mode of the Word document. And we also invited language editors of native English speaker to check language correctness. We have made the following explanations for the questions of the two reviewers.

We hope the changes to meet your requirements for publication.

Thank you for your help.

Yours sincerely,

All authors,

Institute of Entomology

Guizhou University

Response to Reviewer 1 Comments

Point 1: I just only see vitellogenin (Vg) and vitellogenin receptor in abstract, but not in 1. Introduction section, and not in 2.6 Gen expression studies section, I have no idea why test it. So, I suggest add “Expression of genes encoding vitellogenin (Vg) and vitellogenin receptor (VgR) expression” in Introduction.

Response 1: We appreciate your reminder. We have added "Expression of genes encoding vitellogenin (Vg) and Vitellogenin receptor (VgR) Expression" in Introduction according to your comments

Point 2: Table 2,I think it is better to note the gene full name.

Response 2: We have revised according to your comments and added the full name of the gene.

Point 3: Table 5, I think instar is not appropriate, because it including egg, larvae, and pupal. Life stage will be better in column.

Response 3: We appreciate your reminder. Instar has been modified to Life stage

Point 4: Line 263-264, heads/adult is not right.

Response 4: Thank you for your comments. We have reviewed the literature to make corrections.

Point 5: Table 6, please modify the “ Intrinsic rate/(d-1) and Finite rate/d-1)” in the table.

Response 5: We appreciate your reminder. Table has been modified. please see the manuscript.

Point 6: Line 274. “1.16 d- and”

Response 6: Thank you for your comments and we have revised it, please see the manuscript.

Point 7: Line 279 Are there units for expression levels?

Response 7: Thank you for asking is that I didn't make it clear in the manuscript. Relative gene expression was calculated using the 2-△△Ct  quantitative method, so there are no units.

Point 8: Line 231. Change “Treat” into “Treatment”.

Response 8: Thanks for your careful reminding. We have changed it the manuscript.

Point 9: Line 487 Spodoptera frugiperda should be italics

Response 9: Thanks for your careful reminding. We have finished revising the manuscript.

Reviewer 2 Report

Manuscript insects-1926240 by Li et al. describes results of a laboratory toxicology studies of the effects of graphene oxide on fall armyworms. The paper is clearly written. The study was properly designed and executed. Results were straightforward and largely predictable. Discussion is firmly based on available evidence. There are, however, several relatively minor issues that, in my opinion, need to be addressed before publication.

Line 26. Tablet, not table.

Line 95. Spawning usually refers to aquatic vertebrates with external fertilization. I never heard it in reference to insects. I suggest rephrasing it as “after females laid eggs.”

Line 120. Fertility may be a better term than fecundity for the number of eggs hatching. Fecundity then will be limited to the number of eggs laid.

Lines 140-167. It is not clear how many females comprised one sample, and how the study was replicated.

Tables 3 and 4. Which test was used to analyze these data?

Table 5. Which test was used to analyze these data? Why were the means separated among the stages and not among the treatments as in the previous two tables?

Line 264. Offspring, not heads.

Figures 7 and 8. What do the letters above bars indicate?

Lines 422-423. A note of caution about potential damage to the environment should be added. Lines 45-57 provide a lot of information in support of possible non-target effects.

Author Response

Dear Insects Editorial Office

Re: [Insects] Manuscript ID: insects-1926240 - Major Revisions by 7 October 2022

Thank for editors and reviewers’ comments concerning our manuscript entitle ‘Effects of nano-graphene oxide on the growth and reproductive dynamics of Spodoptera frugiperda based on an age-stage, two-sex life table’.

Thank for you and the reviewers’ valuable comments on the improvement of our manuscript. We are very grateful for your detailed instructions on the revision and submission process of our manuscript.

We have carefully considered each comment from the reviewers. We have done the corresponding changes in the 'Revision' mode of the Word document. And we also invited language editors of native English speaker to check language correctness. We have made the following explanations for the questions of the two reviewers.

We hope the changes to meet your requirements for publication.

Thank you for your help.

Yours sincerely,

All authors,

Institute of Entomology

Guizhou University

Response to Reviewer 2 Comments

Point 1: Line 26. Tablet, not table.

Response 1: Thanks again to the reviewers for their very careful review of the article. we have corrected the problems, And it was modified in the manuscript.

Point 2: Line 95. Spawning usually refers to aquatic vertebrates with external fertilization. I never heard it in reference to insects. I suggest rephrasing it as “after females laid eggs.”

Response 2: Based on your comments, we have corrected the problems. please see the manuscript.

Point 3: Line 120. Fertility may be a better term than fecundity for the number of eggs hatching. Fecundity then will be limited to the number of eggs laid.

Response 3: We appreciate your reminder. We have changed it,please see the manuscript.

Point 4: Lines 140-167. It is not clear how many females comprised one sample, and how the study was replicated.

Response 4: . Thank you for your valuable advice. We are sorry for not making it clear.Based on ovarian development and prior grading of FAW [31], female adults that were reared on different concentrations of GO (control, 0.5, 1, 2 mg/g) were sampled at 1, 3, 6, and 11 d after eclosion. Three biological replicates were set for each sampling, and each replicate includes 5 female adults. We have modified them in the manuscript .

Point 5: Tables 3 and 4. Which test was used to analyze these data?

Response 5: Thank you for your careful review and we apologize for not explaining this clearly in the manuscript. Here is the answer to that question: Data are the mean ± standard error. Means in a column followed by different letters are significantly different (Tukey HSD, p < 0.05).

The data was analysedv by method of Tukey HSD. We have added it in the manuscript.

Point 6: Table 5. Which test was used to analyze these data? Why were the means separated among the stages and not among the treatments as in the previous two tables?

Response 6: We appreciate your reminder and we apologize for not explaining this clearly in the manuscript. Here's the answer to that question: The data analysis method is Tukey HSD. Table 5 has been revised to be consistent with the previous two tables. We have revised them in the manuscript.

Point 7: Line 264. Offspring, not heads.

Response 7: Thanks again to the reviewers for their very careful review of the article. We have corrected the problems in the manuscript.

Point 8: Figures 7 and 8. What do the letters above bars indicate?

Response 8: We appreciate your reminder. We apologize for not making this clear in the manuscript. Lowercase letters indicate that the gene expression levels of female Armyworm adults treated with different concentrations of GO at the same time period are significantly different at P<0.05. Capital letters indicate that gene expression levels of female adult Armyworm in the same treatment at different time periods were significantly different at P<0.01. They were revised in the manuscript.

Point 9: Lines 422-423. A note of caution about potential damage to the environment should be added. Lines 45-57 provide a lot of information in support of possible non-target effects.

Response 9: We appreciate your reminder. As there as rarely rereaches on non-target effects of NMs in the field of agriculture, We only mentioned that assessment of NMs risk on non-target organisms is urgently needed. Add the following:

In recent years, numerous studies have been conducted to find out the potential toxicity of graphene family materials (GFMs) to different organisms for assesssing ap-plication prospects of GFMs. As stated by Jastrzebska and Olszyna, considering the in-crease in the use and production of GFMs and the consequent environmental emissions and their toxic effects is becoming an urgent issue [51]. Some studies have shown that GFMs have certain effects on aquatic environment and soil organisms. Xie et al. studied the toxicity of GO to white rot fungus,and found that the growth of white rot fungi could be stimulated at a low concentration of GO, while be inactivated at a high concentration [52]. Li et al. studied the potential adverse effects of graphite, and graphene quantum dots (GQDs) on the motor nervous system using nematode Caenorhabditis elegans and found that graphene-based nanomaterials could cause damages to the dopaminergic and glutamatergic neurons [53]. Chen et al. showed that GO accumula-tions occurred in the liver and intestine and caused obvious chronic toxicity to these organs after 14 days exposure [54]. Although oxicological effects of GFMs to some or-ganisms were determined to a certain extent, much more attention should be paid on more other different organisms. We have added it in the manuscript.

References

[51] Jastrzębska, A. M.; Olszyna A R. The ecotoxicity of graphene family materials: current status, knowledge gaps and future needs. J. Nanopart. Res., 2015, 17(1): 1-21.

[52] Xie, J.; Ming, Z.; Li, H.; Yang, H.; Yu, B.; Wu, R.; Liu, X.; Bai, Y.; Yang, S.T. Toxicity of graphene oxide to white rot fungus Phanerochaete chrysosporium. Chemosphere, 2016, 151: 324-331.

[53] Li, P.; Xu, T.; Wu, S.; Lei, L.; He, D. Chronic exposure to graphene‐based nanomaterials induces behavioral deficits and neural damage in Caenorhabditis elegans. J. Appl. Toxicol. 2017, 37(10): 1140-1150.

[54] Chen, M., Yin, J., Liang, Y., Yuan, S., Wang, F., Song, M., & Wang, H. Oxidative stress and immunotoxicity induced by graphene oxide in zebrafish. Aquat. Toxicol, 2016, 174: 54-60.